# FedDTW: Federated Digital Twin Weighting for Mitigating Client Heterogeneity and Unreliable Connectivity

## Abstract

Federated learning in cross-device settings suffers when selected clients fail to participate, producing biased global updates and slower convergence under partial participation. We introduce Federated Digital Twin Weighting (FedDTW)—a lightweight, server-side mechanism that maintains a digital twin of each client's model to impute missing updates. When a client is unavailable in a round, the server forecasts that client's current parameters from its historical weight trajectory and uses the forecast in aggregation. We evaluate FedDTW under four realistic participation patterns—Random Client Dropout, Variable Participation Rates, Network Partitions, and Delayed Updates—across four time-series datasets (Beijing Air Quality, LTE, Solar Power, METR-LA) and common forecasting backbones (CNN, RNN/GRU/LSTM, DALSTM-AE). FedDTW consistently tracks the full-participation reference (FPR) more closely than FedAvg and yields up to $\approx 6.11$–$50.65\%$ lower RMSE in representative settings. These results indicate that simple, low-parameter weight-forecasting can make FL more resilient to unreliable connectivity without changing client-side training.

## 1 Introduction

Federated Learning (FL), first introduced in McMahan et al. (2017) and applied to various tasks, is an emerging distributed optimization paradigm that enables collaborative training while preserving data privacy by keeping clients' data local. The central server aggregates the local model updates from clients to create a global model, which is then distributed to participants for the next training iteration. Typically, this aggregation is performed by computing a weighted average of the clients' model parameters. However, not all clients can participate in every training round due to various practical challenges, such as network instability, or hardware maintenance, etc. A common approach to addressing this issue is client sampling, wherein only a subset of clients contributes to the global model update at each round. While this reduces the impact of unavailable clients, it also introduces limitations, including missing out on updates from clients with critical datasets and lengthening the convergence time. In fact, numerous sampling strategies have been proposed to mitigate these effects, acknowledging the reality that not all client models are received at every iteration.

Is it possible to ensure the inclusion of all local models in each training round, regardless of network conditions? One practical approach is to keep server-side, continuously updated digital-twin replicas of each client's weights. When a physical device is unavailable, its twin can immediately stand in, with weight forecasting sustaining training continuity despite unstable connectivity. A review of current literature suggests this strategy has not yet been implemented at scale, revealing a gap between the concept of virtual model duplication and empirical methods for handling intermittent participation. Exploring how such an approach could impact model generalization raises several questions: (1) What mechanisms would underpin the optimal coexistence of physical and virtual clients in different FL settings? (2) How would forecasting and aggregating virtual weights affect the learning process? This paper revisits the FL aggregation process under the physical and virtual coexistence paradigm, focusing on forecasting model weights in unstable client participation environments as illustrated in Figure 1 and examining the implications for generalization. The findings provide intriguing insights that open new avenues for federated optimization research.

Interestingly, in cross-device FL, clients' local data is often non-independent and identically distributed (non-IID) due to diverse computational resources Wang et al. (2020), Abdelmoniem et al. (2022), causing client dropouts that really create significant challenges. Obviously, privacy restrictions prevent data sharing, so dropouts often lead to aggregated updates favoring active clients, diverging from the training objective and reducing model effectiveness Ribero et al. (2022), Wang et al. (2021). However, unlike deliberate client sampling, where the server selects accessible clients Yang et al. (2021), Li et al. (2019), Fraboni et al. (2022), unexpected dropouts force reliance on submitted updates, so it results in biased global gradients. In fact, some methods replace missing updates with stored ones Gu et al. (2021), but these can be outdated as the global model evolves.

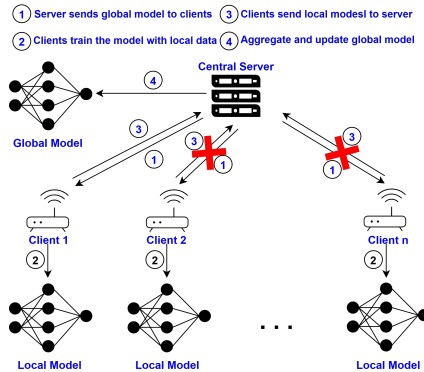

Figure 1: Client dropout example.

Traditionally, FedAvg McMahan et al. (2017) is popularly selected among aggregate methods because of its simplicity and efficiency in performing model aggregation, but its accuracy may oscillate when dealing with fluctuating and high client dropouts. Hence, we come up with a solution to answer the mentioned intriguing question. In brief, our main contributions are summarized as follows:

- We first evaluate the convergence performance of the classical FedAvg algorithm with arbitrary client dropouts on the four scenarios. Theoretical analysis indicates that client dropouts cause a biased update in each training iteration.
- We propose a novel FL algorithm, named FedDTW[1], which is flexible as it is able to work with both IID and non-IID data in addressing the client dropout problems in such scenarios. The core idea is integrating with digital twins to forecast the weights of models whose clients are missing at a specific FL training round based on historical trends.
- We develop a mechanism to correctly extract and manipulate model's weights of each client by applying the simple, yet effective digital twin formulas to forecast missing weights.
- We systematically evaluate FedDTW under realistic client-dropout regimes (random dropout, variable participation, network partitions, delayed updates) across multiple time-series benchmarks and architectures, where it consistently outperforms FedAvg.

## 2 INTEGRATION OF DIGITAL TWINS WITH WEIGHT FORECASTING

For each client $i$, the server maintains a virtual replica that stores historical information about the client's model parameters. When a client cannot provide its model update due to network instability, the server uses the digital twin to forecast the client's current model parameters based on its historical data. This approach aims to mitigate the effects of network instability by providing estimated updates, ensuring that the FL process continues smoothly even when some clients are offline. Let $\theta_t^i$ be the local model parameters of clients $i$ at time $t$. The digital twin stores historical parameters $\{\theta_{t'}^i | t' < t\}$. The server uses a forecasting function $f$ to estimate the current parameters when they are unavailable as $\hat{\theta}_t^i = f(\{\theta_{t'}^i | t' < t\})$, where the estimated parameters $\hat{\theta}_t^i$ are used in place of the missing $\theta_t^i$ during aggregation.

### 2.1 UNSTABLE NETWORK SIMULATION SCENARIOS

Federated clients may intermittently fail to participate in training iterations due to unpredictable events, a phenomenon referred to as client dropout Wang et al. (2020), Abdelmoniem et al. (2022). Consequently, only a portion of clients can complete local training and submit model updates in each iteration, which substantially impairs convergence performance and slows down the training process Imteaj et al. (2021). Figure 2 reflects the real-world client dropouts often happening in FL.

---

[1]The codes are available at https://anonymous.4open.science/r/feddtw-torch-836E

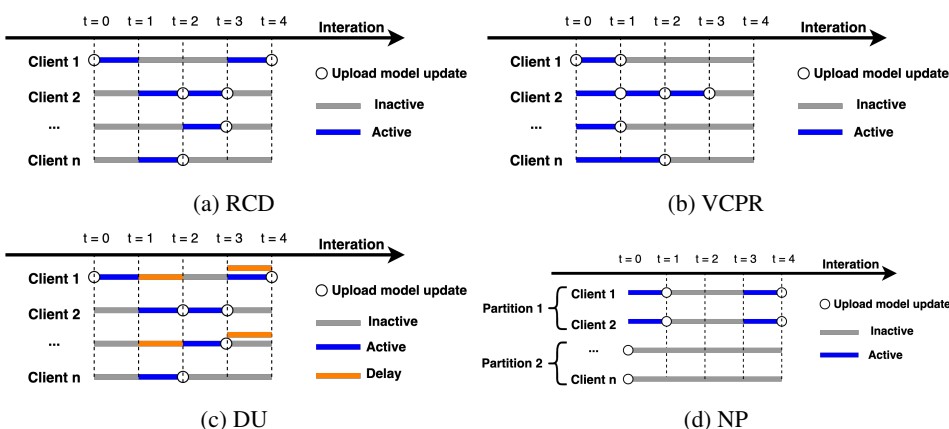

Figure 2: Illustration of the client availability in cross-device FL.

---

**Algorithm 1** Random Client Dropout (RCD)

---

Initialize global model $\theta_0$
Initialize digital twins $\{\theta_0^i\}$ for all clients $i \in \mathcal{C}$
**for** $t = 1$ *to* $T$ **do**
    Randomly select participating clients $\mathcal{S}_t \subseteq \mathcal{C}$
    **foreach** *client* $i \in \mathcal{S}_t$ **do**
        Client $i$ computes update $\theta_t^i$
        Server updates digital twin $\theta_t^i$
    **end**
    **foreach** *client* $i \notin \mathcal{S}_t$ **do**
        Server forecasts $\hat{\theta}_t^i = f(\{\theta_{t'}^i\})$
    **end**
    Aggregate updates:

$$\theta_t = \text{Aggregate}\left(\{\theta_t^i \mid i \in \mathcal{S}_t\} \cup \{\hat{\theta}_t^i \mid i \notin \mathcal{S}_t\}\right)$$

**end**

---

**Algorithm 2** Variable Client Participation Rates (VCPR)

---

Initialize global model $\theta_0$
Initialize digital twins $\{\theta_0^i\}$ for all clients $i \in \mathcal{C}$
**for** $t = 1$ *to* $T$ **do**
    Determine participation probability $p(t)$
    Select participating clients $\mathcal{S}_t$ based on $p(t)$
    **foreach** *client* $i \in \mathcal{S}_t$ **do**
        Client $i$ computes update $\theta_t^i$
        Server updates digital twin $\theta_t^i$
    **end**
    **foreach** *client* $i \notin \mathcal{S}_t$ **do**
        Server forecasts $\hat{\theta}_t^i = f(\{\theta_{t'}^i\})$
    **end**
    Aggregate updates from all clients to obtain $\theta_t$
**end**

---

**Algorithm 3** Network Partitions (NP)

---

Initialize global model $\theta_0$
Initialize digital twins $\{\theta_0^i\}$ for all clients $i \in$ network partitions $\{\mathcal{C}_1, \mathcal{C}_2, \dots\}$
**for** $t = 1$ *to* $T$ **do**
    **foreach** *connected partition* $\mathcal{C}_k$ **do**
        Clients $i \in \mathcal{C}_k$ send updates $\theta_t^i$
        Update digital twins $\theta_t^i$
    **end**
    **foreach** *disconnected partition* $\mathcal{C}_l$ **do**
        **foreach** *client* $i \in \mathcal{C}_l$ **do**
            Forecast $\hat{\theta}_t^i = f(\{\theta_{t'}^i\})$
        **end**
    **end**
    Aggregate updates from all clients to obtain $\theta_t$
**end**

---

**Algorithm 4** Delayed Updates (DU)

---

Initialize global model $\theta_0$
Initialize digital twins $\{\theta_0^i\}$ for all clients $i \in \mathcal{C}$
Define delay schedule for clients
**for** $t = 1$ *to* $T$ **do**
    **foreach** *client* $i$ **do**
        **if** *update* $\theta_t^i$ *is received at time* $t$ **then**
            Update digital twin $\theta_t^i$
        **end**
        **else**
            Forecast $\hat{\theta}_t^i = f(\{\theta_{t'}^i\})$
        **end**
    **end**
    Aggregate updates from all clients to obtain $\theta_t$
**end**

---

In the RCD scenario, clients randomly fail to send updates during certain training rounds. This reflects exactly the real scenario where instances of unpredictable client unavailability, such as hardware failures, temporary network disconnections, or environmental issues preventing updates from clients from being captured. For this reason, the server uses its digital twins to forecast the missing updates for lost clients. We formulate this client dropout case in Algorithm 1.

The VCPR scenario reflects a dynamic and evolving participation pattern over time, which is also very common in most of the FL setups. Suppose that we have client participation rates with $m$ percentage across training rounds, simulating situations where clients face fluctuating availability due to workload variations, resource constraints, or operational priorities. For instance, with an IoT platform, sensors in a monitoring network might take turn to be in active and idle states because of energy-saving protocols or usage schedules. Algorithm 2 illustrates this circumstance.

NP is a setup to ensure the high scalability of a network or prevent physical or logical disruptions such as localized network failures, scheduled maintenance, or natural disasters, it is often partitioned. This results in the isolation of several clients in an FL system. For example, sensors in a specific geographical location may become temporarily disconnected due to maintenance activities or environmental factors. As a result, the server uses digital twins to forecast its updates for clients in disconnected partitions. We formulate the scenario in Algorithm 3.

In the DU scenario, clients often experience delays in sending their updates due to network latency or high traffic at a certain time. For instance, in remote areas or during peak usage periods, clients may struggle to upload their updates to the server promptly. Until the delayed updates arrive, the server uses digital twins to forecast the missing updates. This scenario is described in Algorithm 4.

## 2.2 WEIGHT FORECASTING

In the digital twin environment, where data is insufficient to train complex forecasting models due to dynamic labels, parameter-free or parameter-light models for time series forecasting present suitable alternatives. These methods do not require extensive training data and can be directly applied to forecast missing model weights in federated learning experiments. Before introducing two parameter-free forecasting models, we define the following data representation.

Let $P$ the total number of model parameters (weights) in a model; $T$ the total number of time steps (rounds) in the federated learning process; $\theta_t^i \in \mathbb{R}^P$ the vector of model weights for client $i$ at time $t$; and $\theta_t^{i,p}$ the $p$-th parameter of client $i$ at time $t$. We can represent the historical weights of client $i$ as a matrix $\Theta^i \in \mathbb{R}^{T \times P}$, where each column $p$ represents a time series $\{\theta_t^{i,p}\}_T$ for parameter $p$.

$$\Theta^i = \begin{bmatrix} \theta_1^{i,1} & \theta_1^{i,2} & \cdots & \theta_1^{i,P} \\ \theta_2^{i,1} & \theta_2^{i,2} & \cdots & \theta_2^{i,P} \\ \vdots & \vdots & \ddots & \vdots \\ \theta_T^{i,1} & \theta_T^{i,2} & \cdots & \theta_T^{i,P} \end{bmatrix} \quad (1)$$

### 2.2.1 MOVING AVERAGE FORECASTING (MAF)

The MAF method forecasts the next value in a time series as the average of the most recent $n$ observed values. The MAF method assumes that the future values of the time series are represented by the mean of the most recent past observations. This simple yet effective approach is suitable for forecasting missing model weights in FL setups where minimal computational overhead is desired, especially when the time series lacks clear trends or seasonal patterns. For each parameter $p$ of client $i$, the forecasted weight at time $t$ is given by Equation 2, where $n$ is the window size or average number of past observations. A larger $n$ results in smoother forecasts but may lag behind trends. A smaller $n$ makes the forecast more responsive to recent changes but may be more volatile. $\theta_{t-k}^{i,p}$ is the observed weight at time $t-k$ for parameter $p$.

Note that $n = 2$ in our experiments. Besides, $t \geq n+1$ ensures enough past observations to calculate the moving average. If $t < n+1$, adjust $n$ accordingly to use the available data for the average.

$$\hat{\theta}_t^{i,p} = \frac{1}{n} \sum_{k=1}^{n} \theta_{t-k}^{i,p} \quad (2)$$

### 2.2.2 WEIGHTED SMOOTHING FORECASTING (WSF)

WSF is a recursive forecasting technique where each forecast is a weighted average of the previous observations. The method effectively captures short-term trends, making it suitable for scenarios where recent data points are more relevant than older ones, such as in dynamic environments or FL settings with evolving client models.

For each parameter $p$ of client $i$, the forecasted weight at time $t$ is given in Equation 3:

$$\hat{\theta}_t^{i,p} = \alpha \theta_{t-1}^{i,p} + (1-\alpha)\theta_{t-2}^{i,p} + \Delta\theta \quad (3)$$

where $\alpha \in (0,1)$ is the smoothing factor, which controls the rate at which the influence of past observations decreases. A higher $\alpha$ (closer to 1) places more weight on recent observations, making the forecast more responsive. A lower $\alpha$ (closer to 0) gives more weight to past observations, resulting in smoother forecasts. $\Delta\theta = \theta_{t-1}^{i,p} - \theta_{t-2}^{i,p}$ is an assumption of linear change of weights at a constant rate.

## 3 METHODOLOGY

### 3.1 FL TIME SERIES FORECASTING FORMULATION

Let's consider the problem regarding *individual training* in FL, where each client holds observations and performs its local training. Let $\Omega_t = \{\omega_{t,1}, \cdots, \omega_{t,n}\}$ be the measurements at timestep $t$, with $n$ being the number of variate. For a given $t$, we can look back on a slice of past observations $T \in [t - T + 1, t]$ and $\Omega_t' = \{\Omega_{t-T+1}, \cdots, \Omega_t\}$. In time series forecasting, the main objective is to predict the next measurements based on the past observations (prior lag points) $\Omega_t'$. By utilizing the entire measurements of univariate or multivariate datasets, we can build a model that is capable of generalizing unseen future data. In a FL system, it consists of a central server and $N$ participants (denoted by the set $\mathcal{N} = \{1, \ldots, N\}$), developing a forecasting model designed to generalize to their future observations. These nodes (considered *individual learning*) collaboratively train a model $w \in \mathbb{R}^M$ with $M$ trainable parameters to minimize the loss over data samples of all clients.

The iterative process persists until the global model effectively generalizes across the observations of all $N$ participants with the global training objective expressed in Equation 4.

$$\min_{w \in \mathbb{R}^M} f(w) = \frac{1}{N} \sum_{i=1}^{N} f_i(w) \quad (4)$$

### 3.2 TIME SERIES FORECASTING MODEL SELECTION

We leverage FL to train five popular models, including Convolutional Neural Network (CNN), Gated Recurrent Unit (GRU), Long Short-Term Memory (LSTM), Recurrent Neural Network (RNN), and Dual Attention LSTM Autoencoder (DALSTM-AE) for our time series forecasting experiments. The selection is not for their novelty but because their parameterization is transparent and decomposable. This makes it practical to break down layer- and weight-level trajectories, enabling our server-side digital-twin to forecast missing client updates directly in weight space and to demonstrate FedDTW's mechanism clearly and reproducibly across diverse temporal models. The detailed architectures of these models are thoroughly described in the Appendix A.1 section.

### 3.3 CLIENT DROPOUT SIMULATION MATRIX

We propose a technique that models client presence and absence across $n$ training iterations using a binary matrix with $m$ clients (columns) and $n$ rounds (rows), where *1* indicates participation and *0* denotes dropout. This probably enables two key metrics: *(i)* the percentage of clients absent per round (proportion of zeros in a row), and *(ii)* the percentage of rounds a client is absent (proportion of zeros in a column). By varying the distribution of *1s* and *0s*, we simulate diverse participation scenarios in FL with $m$ clients over $n$ rounds. For the NP setup, we consider a FL system with $m$ total clients grouped into $k$ clusters, each with a fixed number of clients, where only intra-cluster clients contribute to local model updates and aggregation per round. For DU, we introduce a timestamp-based mechanism over a range of periods. For each round $t$ (row index), we compute $t \mod k$ with $k \in [1..9]$. If a client's update falls within the predefined delay range, the server will discard it, even if the client reconnects later, treating a marked present *1* entry as absent *0* for aggregation.

We formulate the client dropout settings with the binary matrix $\mathbf{M}$, illustrated in Equation 5, where $x_r^{c_i} = 1$ indicates the participation at iteration time *r-th* of client *i-th*, and *0* represents an empty update value. Through the matrix, we can reproducibly control the degree of client dropouts as well as find its appropriate smoothing factor $\alpha$.

$$\mathbf{M} = \begin{bmatrix} x_1^{c_1} & x_1^{c_2} & \ldots & x_1^{c_m} \\ 0 & x_2^{c_2} & \ldots & 0 \\ \ldots & \ldots & \ldots & \ldots \\ 0 & x_n^{c_2} & \ldots & x_n^{c_m} \end{bmatrix} \quad (5)$$

## 3.4 FEDERATED TRAINING AND AGGREGATION DESIGN

We follow the traditional FL training process where model aggregation is a critical phase and extend a flexible framework for general time-series forecasting introduced by Perifanis et al. (2023). Algorithm 5 shows the details of the FedAvg and FedDTW aggregate methods when dealing with weight updates under the FPR and dropout scenarios. In addition, the metrics used to assess the local and global model's performance are presented in the Appendix A.3 section.

---

**Algorithm 5** Implementation of $\boxed{\text{FedDTW}}$ and $\boxed{\text{FedAvg}}$ with missing updates

---

**Require:** Local datasets $D^i$, client dropout matrix $M$, number of clients $N$, number of federated rounds $T$, number of local epochs $E$, learning rate $\eta$, smoothing factor $\alpha = 0.8$
**Ensure:** Final global model parameters vector $w^T$
1: Initialize $w^0$
2: **for** $t = 0, 1, ..., T-1$ **do**
3:   Sample a set of parties $S_t$
4:   $n \leftarrow \sum_{i \in S} |D^i|$
5:   **for all** $i \in S$ **in parallel do**
6:     Send the global model $w^t$ to client $C_i$
7:     $\Delta w_i^t, r_i \leftarrow \textbf{LocalTraining}(i, w^t)$
8:   **end for**
9:   $\Delta W \leftarrow \sum_{i \in S} \frac{|D^i|}{n} \Delta w_i^t$
10:   $\Delta ew^t \leftarrow \sum_{i \in \bar{S}} \frac{|D^i|}{n}, \bar{S} = \{i : M_{t,i} = 0\}$ /*excluded dropout clients*/
11:   $\boxed{\text{For FedAvg (under FPR):}}$ $w^{t+1} \leftarrow w^t - \eta \Delta W$
12:   $\boxed{\text{For FedAvg (under dropout):}}$ $w^{t+1} \leftarrow w^t - \eta \Delta W - \Delta ew^t$
13:   $\boxed{\text{For FedDTW (under dropout):}}$ $w^{t+1} \leftarrow w^t - \eta \Delta W - \Delta ew^t + \hat{\theta}_t^{i,p}$ *(defined WSF in Formula 3)*
14: **end for**
15: **return** $w^T$
16: Client executes:
    For every algorithm: $L(w; b) = \sum_{(x,y) \in b} l(w; x, y)$

---

# 4 EXPERIMENTAL SETUP

## 4.1 DATASET DIVERSITY SELECTION

We conducted experiments on the four diverse datasets, each representing non-IID real-world time-series data with distinct temporal and structural properties. The first dataset Perifanis et al. (2023) comprises real multivariate LTE Physical Downlink Control Channel (LTE) measurements collected from three base stations in Barcelona, Spain. This dataset includes eleven features aggregated into two-minute intervals. Similarly, the Beijing Multi-Site Air Quality dataset Chen (2017) provides hourly measurements of air pollutants from 12 nationally controlled monitoring sites. Regarding univariate analysis, we utilized the Solar Power dataset Ilyas et al. (2020), which records power output (in Watts) from 21 solar plants across Aarhus, Denmark, at 5-minute intervals (except between 22:00 and 05:00 daily), reflecting the intermittent nature of renewable energy production. The METR-LA dataset Li et al. (2017), which captures traffic speed data from 50 extracted loop detectors (from total 207 detectors) on Los Angeles County highways, aggregated into 5-minute intervals over four months. These datasets are preprocessed through several primary steps which are thoroughly presented in the Appendix A.4 section.

## 4.2 CLIENT DROPOUT MATRIX CONFIGURATION

We populate the matrix in Equation 5 with varying distributions of *1s* and *0s*, ranging from 10% to 50% overall absence rates in both training iterations and client participation for RCD with uniform random absences, VCPR with skewed or NP intra-cluster dropouts to ensure consistent evaluation of heterogeneity effects across subgroups, along with the DU principle. Basically, higher absence rates generally degrade performance due to incomplete aggregations, but in this paper, we emphasize results at a 50% missing update threshold in the experiment report to accentuate the distinctions.

### 4.3 FL Model training and Smoothing factor selection

The FL model training settings are thoroughly presented in the Appendix A.2 section. Besides, it is worth noting that as depicted in Equation 3 and Algorithm 5, we select two previous weights with the smoothing factor $\alpha = 0.8$ to perform the evaluation for all the experiment scenarios. The reason behind this decision is simply that the most recent updates always preserve the accurate information and distribution trend to carry out the weight forecasting effectively, eliminating staleness. The prediction task involves forecasting the next five measurements using a historical window of $T = 10$, under varying temporal resolutions, and the number of client participation diversity of the datasets.

### 4.4 Results

The performance of FedDTW, compared with FedAvg, as detailed in Table 1, reveals substantial enhancements in predictive accuracy across diverse multivariate datasets and models under the prevalent FL scenarios. Quantitatively, FedDTW demonstrates equal or superior performance to FedAvg in terms of RMSE and MAE across all setups. For instance, in the Beijing Air dataset, FedDTW achieves up to a **6.11%** improvement (up arrow) in RMSE and **5.56%** in MAE. Similarly, in the LTE dataset, FedDTW improves more than **4.55%** in RMSE and **8.33%** in MAE. Obviously, these improvements demonstrate the simple yet effective digital twin formulas in combination with the appropriate smoothing factor selection, where the most recent updates preserve the accurate information and distribution trend for the virtual weight replicas, ensuring the continuity of FL forecasting.

Table 1: Averaged RMSE and MAE of the global model on the multivariate datasets

| | Model | | CNN | | LSTM | | GRU | | DALSTM-AE | |
|---|---|---|---|---|---|---|---|---|---|---|
| **Dataset** | **Method** | **Setup** | RMSE | MAE | RMSE | MAE | RMSE | MAE | RMSE | MAE |
| | | RCD | 0.122 | 0.101 | 0.122 | 0.101 | 0.123 | 0.102 | 0.122 | 0.101 |
| | FPR | VCPR | 0.122 | 0.101 | 0.122 | 0.101 | 0.123 | 0.102 | 0.122 | 0.101 |
| | | DU | 0.122 | 0.101 | 0.122 | 0.101 | 0.123 | 0.102 | 0.122 | 0.101 |
| | | NP | 0.122 | 0.101 | 0.122 | 0.101 | 0.123 | 0.102 | 0.122 | 0.101 |
| | | RCD | **0.123**↑ | **0.102**↑ | **0.123**↑ | **0.102**↑ | **0.123**↑ | **0.102**↑ | **0.123**↑ | **0.102**↑ |
| **Beijing Air** | **FedDTW** | VCPR | **0.128**↑ | **0.105**↑ | **0.123**↑ | **0.101**↑ | **0.123**↑ | **0.101**↑ | **0.123**↑ | **0.101**↑ |
| | | DU | **0.124**↑ | **0.102**↑ | **0.123**↑ | **0.101**↑ | **0.123**↑ | **0.102**↑ | **0.123**↑ | **0.101**↑ |
| | | NP | **0.123**↑ | **0.102**↑ | **0.123**↑ | **0.101**↑ | **0.123**↑ | **0.101**↑ | **0.123**↑ | **0.101**↑ |
| | | RCD | 0.124 | 0.103 | 0.124 | 0.102 | 0.124 | 0.103 | 0.124 | 0.102 |
| | FedAvg | VCPR | 0.132 | 0.108 | 0.131 | 0.107 | 0.131 | 0.107 | 0.131 | 0.107 |
| | | DU | 0.132 | 0.108 | 0.131 | 0.107 | 0.131 | 0.107 | 0.131 | 0.107 |
| | | NP | 0.126 | 0.104 | 0.126 | 0.103 | 0.126 | 0.103 | 0.126 | 0.103 |
| | | RCD | 0.023 | 0.011 | 0.021 | 0.011 | 0.022 | 0.011 | 0.023 | 0.012 |
| | FPR | VCPR | 0.023 | 0.011 | 0.021 | 0.011 | 0.022 | 0.011 | 0.023 | 0.012 |
| | | DU | 0.023 | 0.011 | 0.021 | 0.011 | 0.022 | 0.011 | 0.023 | 0.012 |
| | | RCD | **0.023**↕ | **0.011**↑ | **0.021**↑ | **0.011**↕ | **0.022**↕ | **0.011**↑ | 0.025 | **0.013**↕ |
| **LTE** | **FedDTW** | VCPR | **0.023**↕ | **0.012**↕ | **0.022**↕ | **0.011**↕ | **0.022**↕ | **0.011**↕ | **0.025**↕ | **0.013**↕ |
| | | DU | **0.022**↑ | **0.011**↑ | **0.021**↑ | **0.011**↕ | **0.022**↕ | **0.011**↕ | **0.025**↕ | **0.013**↕ |
| | | RCD | 0.023 | 0.012 | 0.022 | 0.011 | 0.022 | 0.011 | 0.024 | 0.013 |
| | FedAvg | VCPR | 0.023 | 0.012 | 0.022 | 0.011 | 0.022 | 0.012 | 0.025 | 0.013 |
| | | DU | 0.023 | 0.012 | 0.022 | 0.011 | 0.022 | 0.012 | 0.025 | 0.013 |

Similarly, with the univariate Solar Power and METR-LA datasets, the average RMSE and MAE values are reported in Table 2 where FedDTW also consistently outperforms FedAvg. FedDTW achieves RMSE and MAE reductions of up to **50.65%** and **46.58%**, respectively, with the Solar Power dataset, while the improvement is up to **37.69%** in both RMSE and MAE with the METR-LA. It is worth noting that we replace DALSTM-AE with RNN model for univariate datasets because of aligning model complexity with data characteristics to provide a balanced comparison. The details of extensive experiments for the univariate datasets are presented in the Appendix A.5 section.

The NRMSE trends illustrated in Figures 3-6 provide deeper insights into FedDTW's performance dynamics over 100 rounds, with blue lines (FedDTW) consistently positioned below red lines (FedAvg) across all models and scenarios. This consistent superiority is particularly pronounced in scenarios with higher client interdependencies and communication challenges, where the gap between FedDTW and FedAvg widens. This trend suggests that as the number of clients increases or the complexity of dependencies grows, our solution's ideal performance becomes more apparent, likely due to its ability to adapt to heterogeneous data distributions and communication disruptions.

In terms of convergence behavior, FedDTW exhibits superior stability and efficiency compared to FedAvg, a result of its innovative use of historical trending data to forecast missing weights. By predicting missing weight updates, FedDTW approximates the performance of the FPR scenario,

leading to faster convergence and reduced error accumulation over training rounds. Moreover, the integration of lightweight digital twin formulas for this forecasting process minimizes computational overhead, ensuring that FedDTW imposes no significant burden on the FL training pipeline. Note that, we decided not to visualize some first epochs since their significant improvement makes the subsequent discrepancy between FedAvg and FedDTW unclear, as illustrated in the Appendix A.6 section at Figures 7-10 with complete epoch visualization.

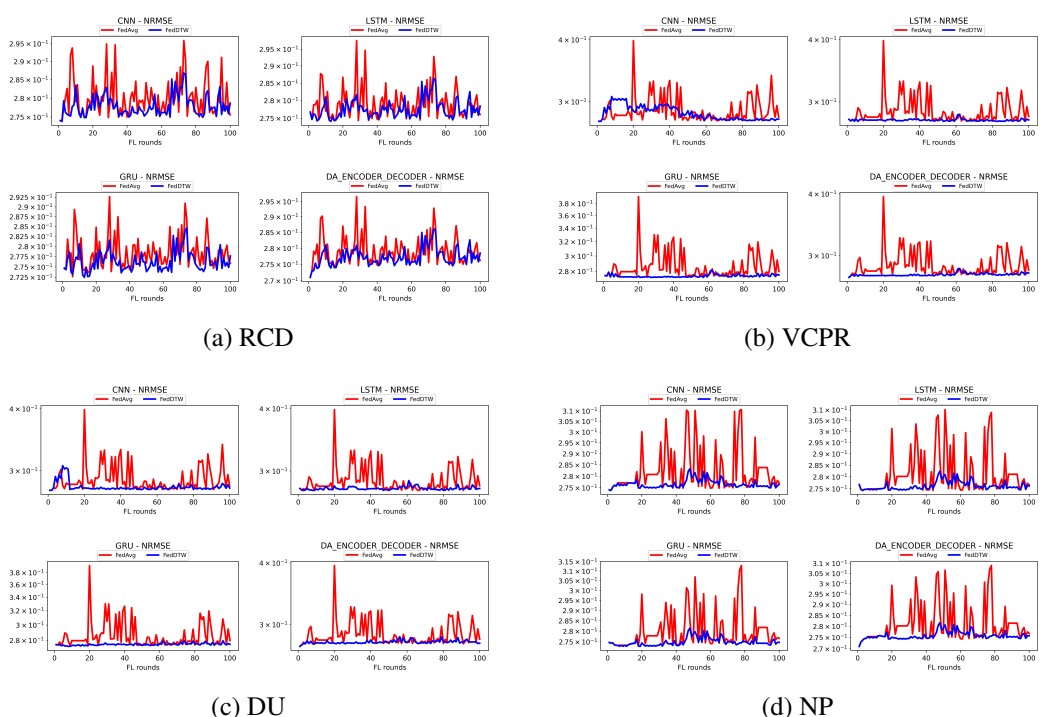

Figure 3: Global NRMSE across scenarios and models on Beijing Air dataset.

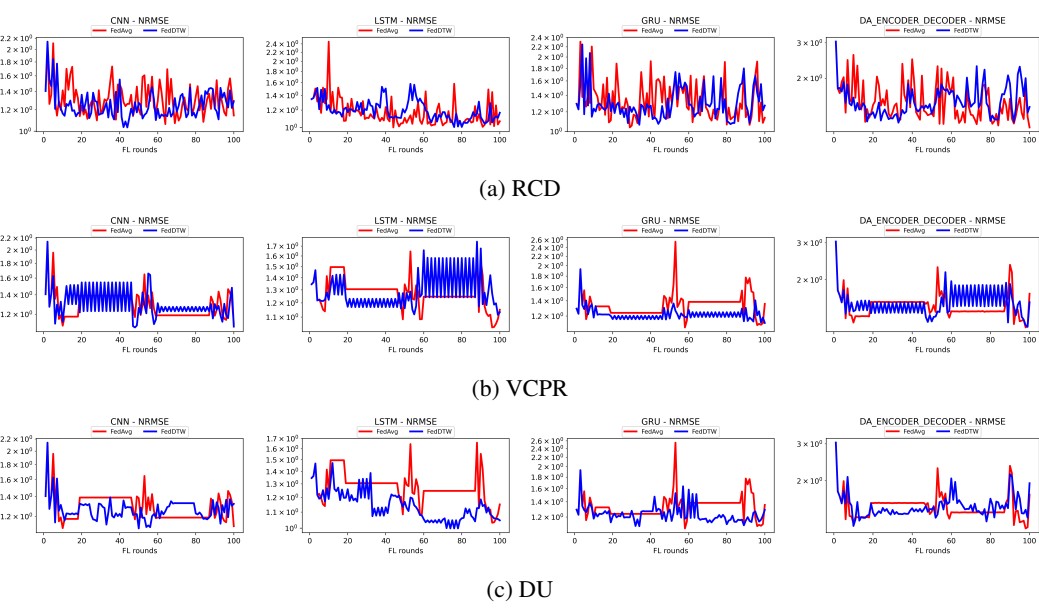

Figure 4: Global NRMSE across scenarios and models on LTE dataset.

## 5 RELATED WORK

FedAvg McMahan et al. (2017) is a widely adopted FL algorithm due to its simplicity, communication efficiency, and strong performance under IID conditions, serving as a baseline in early FL applications Kairouz et al. (2021). However, its performance degrades significantly under non-IID data and partial client participation Zhao et al. (2018), yet it remained prevalent due to the lack of robust alternatives initially. While frameworks such as Flower Beutel et al. (2020) support basic parameter aggregation and fault-tolerant strategies like FedAvg to mitigate dropout effects, they still lack advanced weight extraction and manipulation for forecasting updates from unavailable clients.

Mimic Sun et al. (2023) mitigates dropouts by aligning local updates with central updates through correction values but lacks weight forecasting, focusing on mimicking centralized behavior without predicting absent clients' contributions. The challenge of client heterogeneity and intermittent availability has been studied from various perspectives Zhang et al. (2022). Ribero et al. Ribero et al. (2022) introduced F3AST, an algorithm adapting to client availability patterns, achieving up to 186% accuracy improvements over FedAvg on CIFAR100. Yan et al. Yan et al. (2024b) proposed FedLaAvg, leveraging gradients from all clients for stable training across convex and non-convex settings. Besides, Jhunjhunwala et al. Jhunjhunwala et al. (2022) addressed participation variance with FedVARP, which stores recent client updates as proxies for non-participating clients. Their theoretical analysis shows FedVARP eliminates error due to partial participation without additional computation costs. Similarly, Wang and Ji Wang & Ji (2022) developed a unified analysis framework demonstrating that under specific conditions, FL algorithms with arbitrary participation can achieve convergence rates matching idealized scenarios. For client unavailability, Jiang et al. Jiang et al. (2024) introduced FedAR with local update approximation and rectification, improving test accuracy by up to 75% compared to baselines. Moreover, Rodio and Neglia Rodio & Neglia (2024) proposed FedStale, leveraging stale updates through staleness-aware weighting mechanisms. The impact of biased client selection was examined by Cho et al. Cho et al. (2022), showing it can shift convergence points to favor frequently selected clients. In the same vain, Mitra et al. Mitra et al. (2021) developed FedLin, achieving linear convergence despite heterogeneity through gradient correction and client-specific learning rates. While these approaches address different aspects of heterogeneity, they mainly focus on client selection or retroactive utilization of existing updates rather than predictive modeling of client behavior, lack predictive capabilities for future model states.

Digital twins' application in FL systems remains largely unexplored. Yan et al. Yan et al. (2024a) developed a V2V-enhanced algorithm considering energy constraints and mobility patterns, improving classification accuracy by 3.18% on CIFAR-10. Chahoud et al. Chahoud et al. (2023) introduced an on-demand client deployment framework using containerization for dynamic environments, while Liu et al. Liu et al. (2022) explored client selection in 5G/B5G networks. Weight forecasting in FL contexts remains relatively unexplored. FedVARP and FedAR implicitly use simple prediction by reusing recent updates, but do not model temporal evolution of weights. Momentum-based techniques typically focus on accelerating convergence rather than directly addressing unavailability.

FedDTW bridges these research areas by leveraging digital twin technology with weight forecasting for continuous client participation. While FedVARP, FedAR, and FedStale utilize previous updates, they lack predictive capabilities for future model states. It actively anticipates client model evolution, enabling accurate virtual representations even during extended periods of unavailability. This predictive approach represents a novel direction in FL with significant implications for system resilience in real-world deployments.

## 6 CONCLUSION AND FUTURE WORK

We presented FedDTW, a novel solution designed to address client dropout problems, leading to missing weight updates in FL through the integration of digital twin-based weight forecasting mechanisms, a capability not currently addressed by existing frameworks or libraries. FedDTW consistently outperforms FedAvg, achieving results near the FPR scenarios. Its ability to maintain high-quality global models despite intermittent updates and dynamic participation underscores its reliability, making it a vital solution for applications in various areas. Future work will explore its adaptability to additional areas and envision extending its capabilities within standard FL platforms.

REPRODUCIBILITY STATEMENT

The data and analysis code used to generate the results presented in this paper are available at this repository: https://anonymous.4open.science/r/feddtw-torch-836E under an open-source license. The experiment was performed using Python (version 3.10) with common and machine learning libraries on an Ubuntu Linux operating system. The complete computational environment and detailed instructions for reproducing the analysis are provided in the repository's README file.

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

# A  APPENDIX

In this section, we provide supplementary data, figures, and detailed analyses that complement the findings outlined in the manuscript. To be more precise, this part offers in-depth insights into the model architectures, evaluation metrics, data preprocessing techniques, FL training parameters, experimental reports, as well as performance outcomes across the univariate datasets.

## A.1  MODELS AND LEARNING SETTING

We performed comprehensive experiments on the provided datasets to assess the efficacy of deep learning models within the designed FL framework. The following are the architectural details:

*RNN*: A standard Recurrent Neural Network with a single layer of 128 units, followed by a MLP with one hidden layer of 128 units.

*LSTM*: An enhanced version of RNN, designed to mitigate exploding gradient issues and handle extended sequential data Lara-Benítez et al. (2021). It includes a 128-unit LSTM layer, with its output connected to a MLP featuring one 128-unit hidden layer.

*GRU*: Similar to LSTM, this model addresses exploding gradients in RNNs but with a reduced parameter set for computational efficiency. It comprises a 128-unit GRU layer, followed by a MLP with a single 128-unit layer.

*CNN*: This network processes raw data directly using convolutional layers. The chosen CNN accepts a three-dimensional input matrix of size $(1, T, \#variables)$ and applies four 2D convolutional layers with filter sizes $\{16, 16, 32, 32\}$. The output is passed through a 2D average pooling layer and then to a fully-connected layer with 128 units.

*DALSTM-AE*: This model integrates LSTM and autoencoder (AE) networks into an encoder-decoder LSTM architecture that captures the long-term temporal features. Dual attention module is introduced to enhance the decoder's ability to capture different dynamic features of variables, which can effectively solve the information loss problem induced by overly complex and long sequences. DALSTM-AE includes a 64-unit LSTM layer and 64-unit encoder and decoder hidden layers.

## A.2  MODEL TRAINING AND OPTIMIZATION

We employed the Adam optimizer Kinga et al. (2015) with a learning rate of 0.001, utilizing ReLU as the activation function across layers. Training was optimized using Mean Squared Error (MSE) with a batch size of 128. Besides, we conducted 100 rounds with 5 local epochs per participant with the subsequent data preprocessing tasks and no client sampling was performed. The experiments were executed on a workstation equipped with CPU Intel Xeon Gold 5117, 2048GB of memory and an NVIDIA A40 48GB GPU, using Python 3.10. For each experiment, we report averaged results

and stability metrics, including RMSE, MAE, NRMSE, training and validation losses, obtained by retraining the models from scratch with different random seeds.

## A.3 EVALUATION SCENARIOS AND METRICS

We conduct experiments on each dataset with the real-world challenging scenarios: RCD, VCPR, NP and DU to assess the robustness of the FedAvg and FedDTW across diverse FL conditions. We assess model performance using Mean Absolute Error (MAE), Root Mean Squared Error (RMSE), and Normalized RMSE (NRMSE) as described below:

$$MAE = \frac{1}{n} \sum_{i=1}^{n} |\hat{y}_i - y_i|, \quad RMSE = \sqrt{\frac{\sum_{i=1}^{n} (\hat{y}_i - y_i)^2}{n}}, \quad NRMSE = \frac{1}{\bar{y}} \sqrt{\frac{\sum_{i=1}^{n} (\hat{y}_i - y_i)^2}{n}} \quad (6)$$

In the specified metrics, $n$ is the number of observed values, $y_i$ is the real measurement at $i-th$ and $\hat{y}_i$ is its corresponding prediction.

## A.4 DATA PREPROCESSING

We conduct experiments on four datasets in which the **Beijing Multi-Site Air Quality** Chen (2017) and the real **LTE Physical Downlink Control Channel (LTE)** Perifanis et al. (2023) are multivariate datasets while the public traffic network **METR-LA** Li et al. (2017) and the **Solar Power** Ilyas et al. (2020) datasets are univariate. Our preprocessing pipeline comprises five primary steps:

*Non-IID data introducing.* In fact, these datasets are typically balanced among clients, so we intentionally introduced non-IID data for each client by utilizing the methodology Maat et al. (2017) for generating synthetic time series. The reason of this synthetic data introduction is because of the performance degradation of FedAvg on such data. Although the authors in McMahan et al. (2017) claim that FedAvg can handle non-IID data to to a certain degree, numerous studies Zhao et al. (2018) indicate that accuracy in FL typically declines with non-IID or heterogeneous data. In general, this performance drop is largely due to weight divergence in local models caused by non-IID data. Specifically, local models with identical initial parameters diverge due to varying local data distributions. In FL, the gap between the shared global model, formed by averaging local models, and the ideal model (trained on IID data) widens over time, slowing convergence and degrading learning performance. Particularly, in the context where problematic clients cannot update their weights timely, the weakness of FedAvg can be significantly worse.

*Data cleansing.* This preprocessing step addresses missing or corrupted data and manages outliers. We employ a straightforward approach by replacing missing values with zeros and applying flooring and capping techniques to handle outliers. Zero transformation is preferred over removal to maintain data continuity. Additionally, imputing missing values with a constant may not accurately represent time-series data, while estimating them can be computationally and energetically costly.

*Data split.* The data are divided into three subsets for model training, evaluation and testing. Specifically, the data are split into 60% for training, 20% for validation and 20% for testing.

*Data scaling.* Min-Max normalization is used to eliminate the influence of value ranges.

*Time-series representation.* After applying the above steps, the data are represented as time-series using a sliding window of $T$. Note that we use ten previous values (lag) to predict the next data.

## A.5 PERFORMANCE ON UNIVARIATE DATASETS

Table 2 provides a comprehensive performance comparison across univariate time series datasets under diverse FL setups with RMSE and MAE evaluation for recurrent and convolutional models. The proposed FedDTW framework consistently outperforms the conventional FedAvg approach, achieving RMSE and MAE reductions of up to **50.65%** and **46.58%**, respectively, with the Solar Power dataset, and up to **37.69%** in both RMSE and MAE with the METR-LA dataset. Undoubtedly, this can be again attributable to its innovative incorporation of lightweight digital twin mechanisms that capture historical weight trends of individual clients to forecast missing weights. By aligning sequence alignments and approximating absent clients' impacts, FedDTW preserves the accuracy of the global model, relatively comparable with the FPR, while its computational efficiency ensures

scalability with minimal overhead. This approach is particularly critical in scenarios with successive missing updates such as VCPR, NP and DU, where FedAvg's oversight could trigger cascading performance degradation, thus highlighting FedDTW's efficacy in resource-constrained, large-scale systems.

Table 2: Averaged RMSE and MAE of the global model on the univariate datasets

| Model | | | CNN | | LSTM | | GRU | | RNN | |
|---|---|---|---|---|---|---|---|---|---|---|
| Dataset | Method | Setup | RMSE | MAE | RMSE | MAE | RMSE | MAE | RMSE | MAE |
| Solar Power | FPR | RCD | 0.036 | 0.036 | 0.034 | 0.034 | 0.034 | 0.033 | 0.034 | 0.034 |
| | | VCPR | 0.036 | 0.036 | 0.034 | 0.034 | 0.034 | 0.033 | 0.034 | 0.034 |
| | | DU | 0.036 | 0.036 | 0.034 | 0.034 | 0.034 | 0.033 | 0.034 | 0.034 |
| | | NP | 0.036 | 0.036 | 0.034 | 0.034 | 0.034 | 0.033 | 0.034 | 0.034 |
| | FedDTW | RCD | 0.041 | 0.041 | 0.040 | 0.039 | 0.040 | 0.040 | 0.040 | 0.040 |
| | | VCPR | **0.041↑** | **0.041↑** | **0.045↑** | **0.044↑** | **0.047↑** | **0.047↑** | **0.045↑** | **0.045↑** |
| | | DU | **0.042↑** | **0.041↑** | **0.038↑** | **0.038↑** | **0.040↑** | **0.040↑** | **0.040↑** | **0.039↑** |
| | | NP | **0.038↑** | **0.038↑** | **0.043↑** | **0.043↑** | **0.046↑** | **0.046↑** | **0.046↑** | **0.046↑** |
| | FedAvg | RCD | 0.038 | 0.038 | 0.036 | 0.036 | 0.035 | 0.035 | 0.036 | 0.036 |
| | | VCPR | 0.072 | 0.072 | 0.070 | 0.070 | 0.069 | 0.069 | 0.073 | 0.073 |
| | | DU | 0.072 | 0.072 | 0.070 | 0.070 | 0.069 | 0.069 | 0.073 | 0.073 |
| | | NP | 0.077 | 0.077 | 0.075 | 0.074 | 0.073 | 0.073 | 0.080 | 0.080 |
| METR-LA | FPR | RCD | 0.089 | 0.089 | 0.085 | 0.085 | 0.081 | 0.080 | 0.084 | 0.084 |
| | | VCPR | 0.089 | 0.089 | 0.085 | 0.085 | 0.081 | 0.080 | 0.084 | 0.084 |
| | | DU | 0.089 | 0.089 | 0.085 | 0.085 | 0.081 | 0.080 | 0.084 | 0.084 |
| | | NP | 0.089 | 0.089 | 0.085 | 0.085 | 0.081 | 0.080 | 0.084 | 0.084 |
| | FedDTW | RCD | 0.092 | 0.092 | **0.087↕** | 0.087 | **0.082↕** | **0.082↕** | **0.086↕** | **0.086↕** |
| | | VCPR | **0.083↑** | **0.083↑** | **0.087↑** | **0.087↑** | **0.082↑** | **0.082↑** | **0.085↑** | **0.085↑** |
| | | DU | **0.091↑** | **0.091↑** | **0.084↑** | **0.084↑** | **0.081↑** | **0.081↑** | **0.084↑** | **0.084↑** |
| | | NP | **0.081↑** | **0.081↑** | **0.088↑** | **0.088↑** | **0.083↑** | **0.083↑** | **0.084↑** | **0.084↑** |
| | FedAvg | RCD | 0.091 | 0.091 | 0.087 | 0.086 | 0.082 | 0.082 | 0.086 | 0.086 |
| | | VCPR | 0.122 | 0.122 | 0.115 | 0.115 | 0.107 | 0.107 | 0.113 | 0.113 |
| | | DU | 0.122 | 0.122 | 0.115 | 0.115 | 0.107 | 0.107 | 0.113 | 0.113 |
| | | NP | 0.130 | 0.130 | 0.120 | 0.120 | 0.114 | 0.114 | 0.119 | 0.119 |

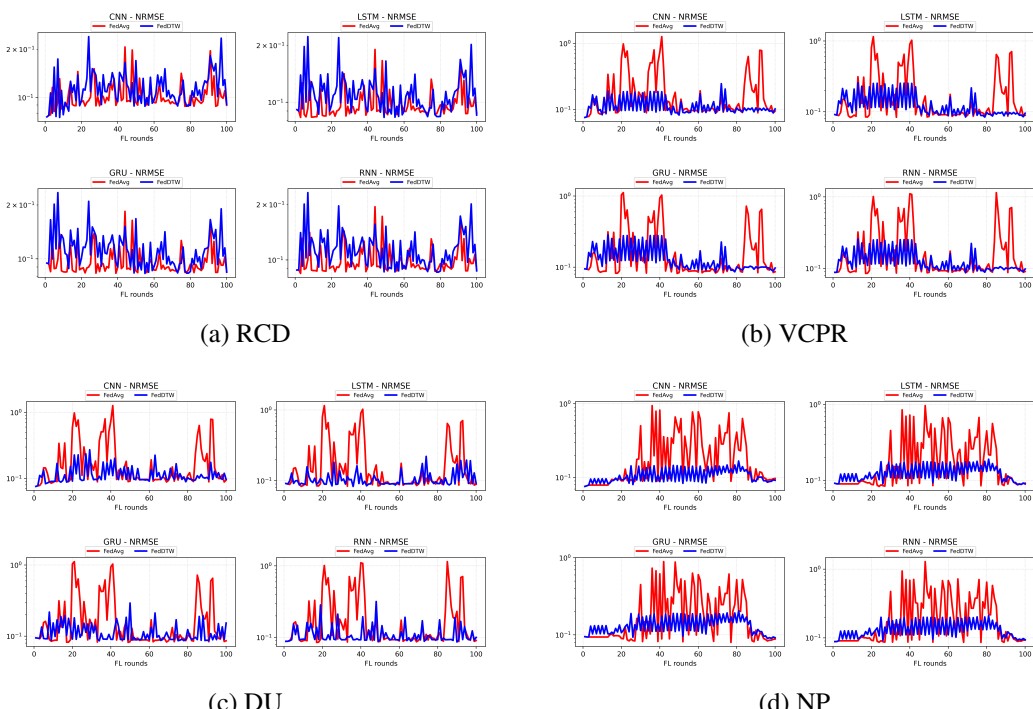

(a) RCD

(b) VCPR

(c) DU

(d) NP

Figure 5: Global NRMSE across scenarios and models on Solar Power dataset.

## A.6 TRAINING LOSS OVERVIEW ACROSS MODELS AND DATASETS

The comparative analysis of global test loss across 100 FL training iterations, as depicted in Figures 7-10, provides a comprehensive insight about the loss during the model training. For the LTE dataset, which comprises only three clients, the performance advantage of FedDTW over FedAvg is less visually pronounced due to the limited client diversity and scale. Nevertheless, FedDTW consistently exhibits lower test loss values compared to FedAvg across all models and scenarios, with the

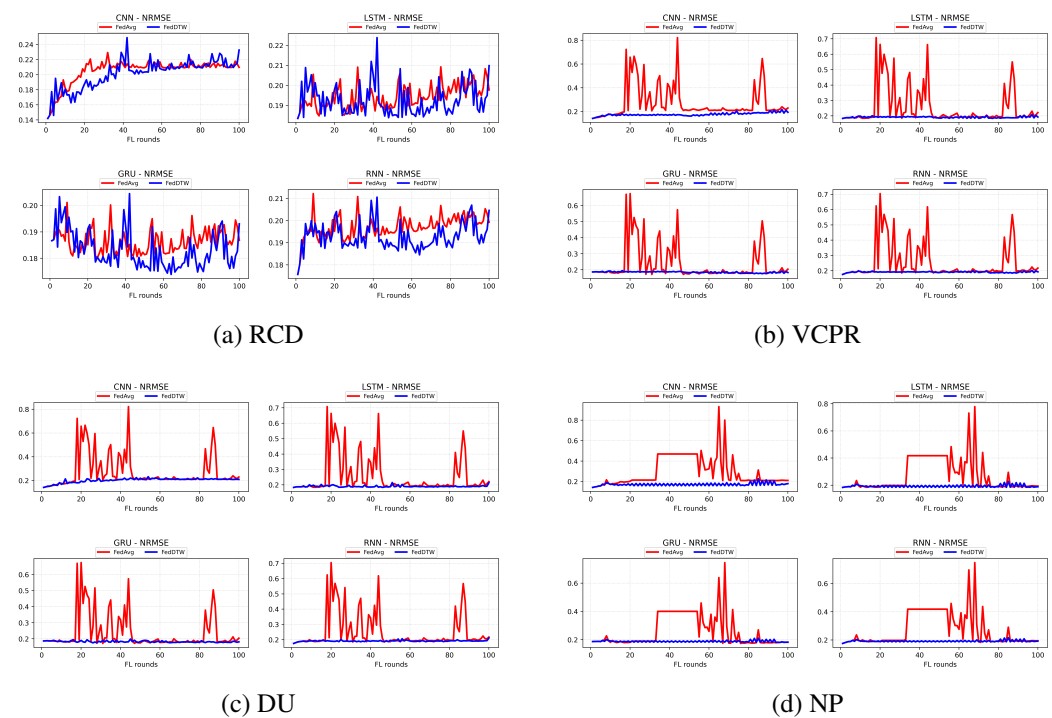

Figure 6: Global NRMSE across scenarios and models on METR-LA dataset

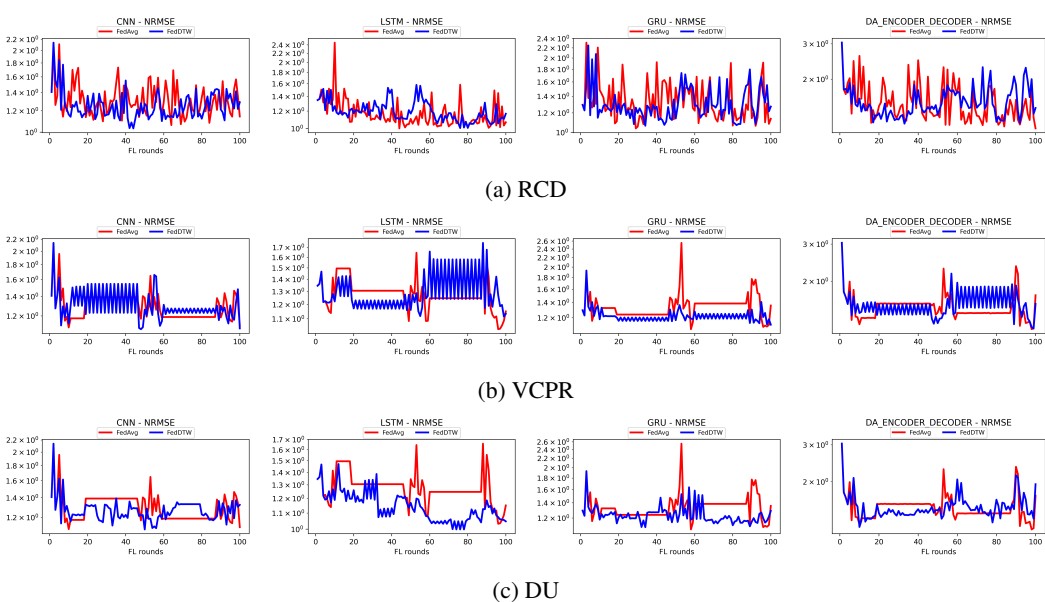

Figure 7: Global test loss across models on LTE dataset.

difference becoming more apparent during rounds with missing client updates. This suggests that even with a small number of clients, FedDTW's ability to mitigate the impact of missing updates provides a marginal but consistent improvement over FedAvg, which tends to overlook these gaps, potentially leading to suboptimal global model updates. In contrast, the remaining datasets having 12, 21, and 50 clients respectively, amplifies the superiority of FedDTW over FedAvg, particularly under scenarios with more complex update patterns.

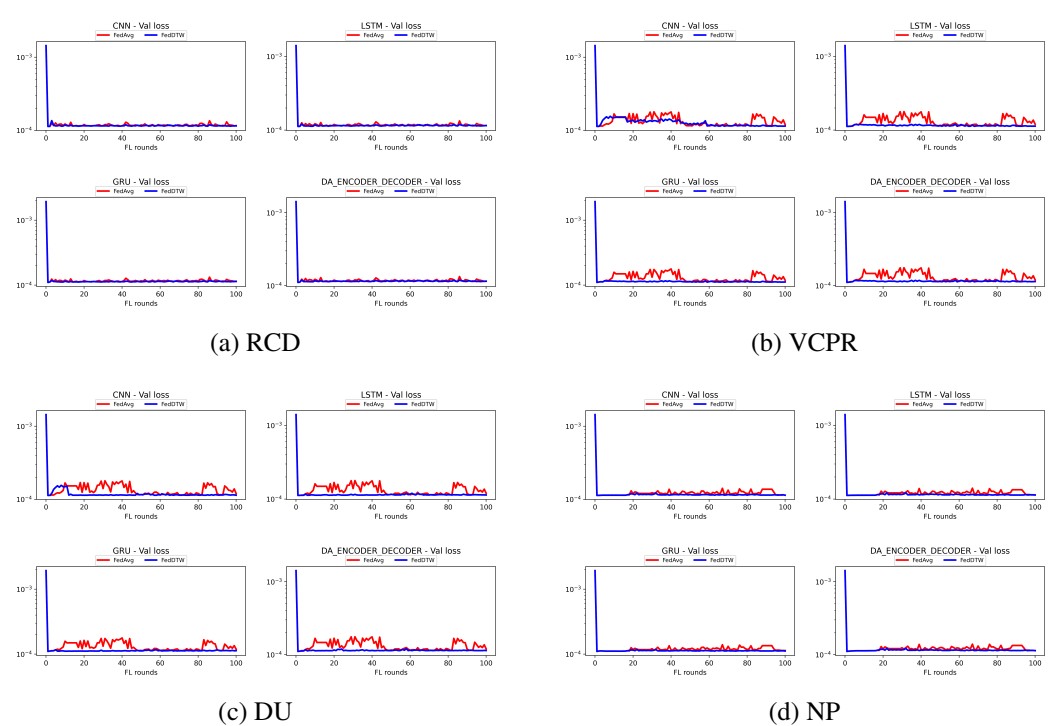

Figure 8: Global test loss across scenarios and models on Beijing Weather dataset.

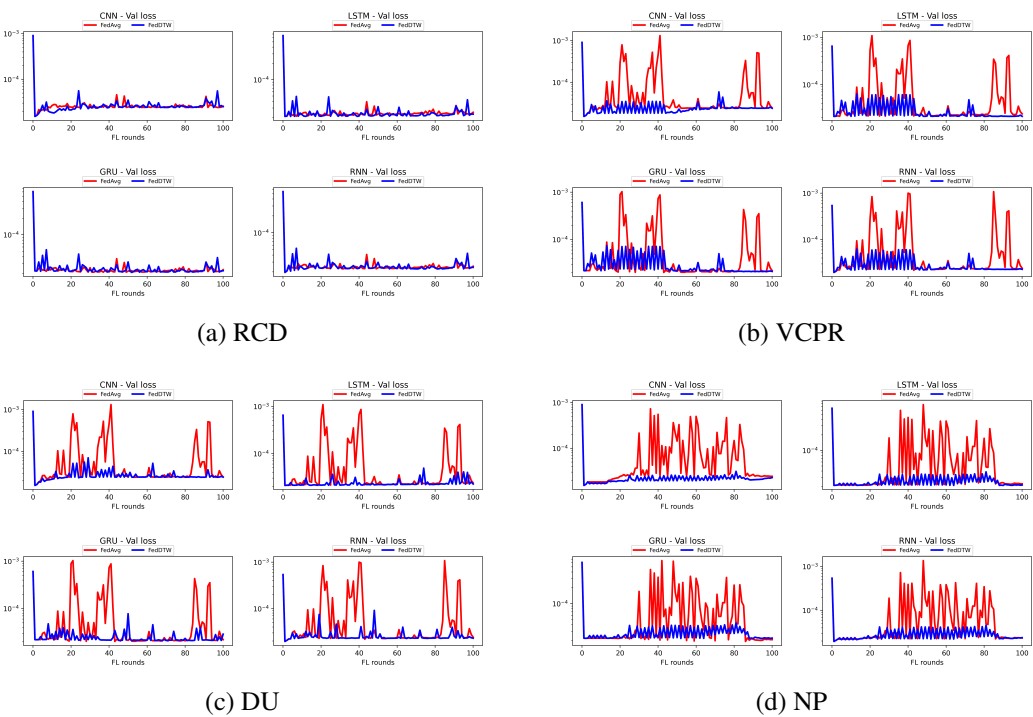

Figure 9: Global test loss across models on Solar Power dataset

Indeed, in the RCD scenario, where client updates are randomly missing, the test loss curves for FedDTW and FedAvg show a relatively modest separation, reflecting the intermittent nature of dropouts that does not consistently disrupt the training process across multiple rounds. However,

in the VCPR, NP, and DU scenarios, where missing updates occur across successive rounds, the performance gap widens significantly. Obviously, in the these setups, the test loss of FedAvg exhibits pronounced spikes and instability, indicative of its failure to adapt to prolonged absences of client contributions, whereas FedDTW maintains a smoother and lower loss trajectory.

Note that due to the constraint of only three clients in the LTE dataset, its perfect condition supports us in evaluating the behavior and performance of our approach on a limited FL client participation system. Therefore, we do not necessarily experiment and provide the assessment report for this dataset under the NP client dropout scenario.

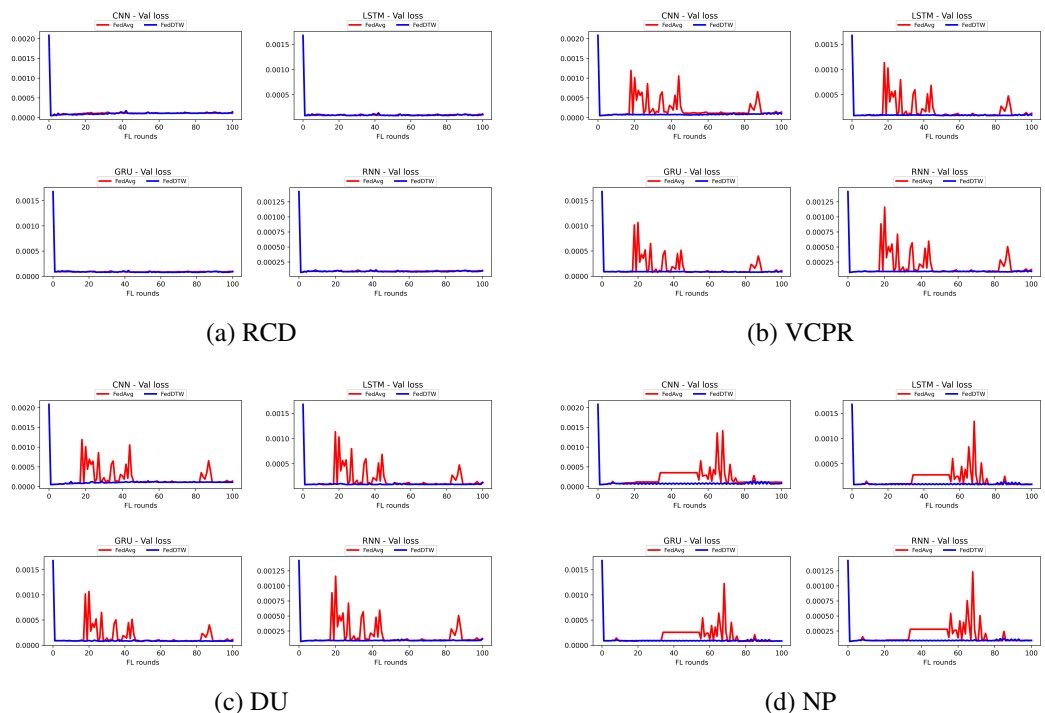

Figure 10: Global test loss across models on METR-LA dataset.

In conclusion, these results underscore FedDTW's potential as a superior alternative to FedAvg, particularly in environments with unreliable client participation or communication, and suggest avenues for future research into optimizing digital twin forecasting for even greater resilience across varying client dynamics.

