# OpenReview forum: "FedDTW: Federated Digital Twin Weighting for Mitigating Client Heterogeneity and Unreliable Connectivity"
_ICLR.cc/2026/Conference — ICLR 2026 Conference Withdrawn Submission_

### Official Review · Reviewer_T1PG · 2025-10-16

**Soundness:** 1
**Presentation:** 2
**Contribution:** 1
**Rating:** 2
**Confidence:** 5

**Summary:**

This paper introduces FedDTW (Federated Digital Twin Weighting), a lightweight, server-side algorithm that mitigates the impact of client dropouts. The idea of FedDTW is to maintain a virtual replica of each client's parameter update on the server. When a client is unavailable, FedDTW forecasts its model update based on its historical weight trajectory and uses this forecast in the global aggregation.

**Strengths:**

FedDTW significantly outperforms FedAvg in the case of client dropout.

**Weaknesses:**

1.The authors claim that this paper first evaluates the convergence of FedAvg with client dropout, which is mistaken. The convergence analysis of FedAvg under client dropout has already been proposed in [1].

2.The authors state that “Theoretical analysis indicates that client dropouts cause a
biased update in each training iteration” in the introduction, while I cannot find the theoretical analysis after searching through the paper.

3.This paper lacks proper citation. I believe that the ideas of Moving Average Forecasting (2.2.1) and Weighted Smooth Forecasting (2.2.2) are not original in this paper. However, the authors haven’t provided any citation regarding these two modules.

4.The authors only compare FedDTW with FedAvg in the experiment, while more state-of-the-art methods addressing client dropout should be included, such as [1-3].

5.It is unclear what model architectures are used for clients’ local models.

6.The tasks and datasets used in this paper’s experiment are quite simple. More deep-learning datasets (like cifar10 and cifar100) and models (e.g. ResNet) should be implemented.

7.It is unclear how to determine the historical window $n$ in section 3.3.

8.The idea of applying exisitng techniques (moving average forecasting and weighted smooth forecasting) to forecast client update is too simple and lacks novelty.

[1] Fast Federated Learning in the Presence of Arbitrary Device Unavailability, NeurIPS 2021.

[2] Efficient Federated Learning against Heterogeneous and Non-stationary Client Unavailability, NeurIPS 2024.

[3] Mitigating Persistent Client Dropout in Asynchronous Decentralized Federated Learning, KDD 2025.

**Questions:**

Please refer to the weaknesses.

---

### Official Review · Reviewer_dNq3 · 2025-10-21

**Soundness:** 1
**Presentation:** 2
**Contribution:** 1
**Rating:** 2
**Confidence:** 5

**Summary:**

This paper explores a digital twin-based federated learning scheme, *FedDTW*, that aims to enable reliable global model training in realistic environments where devices may be disconnected. The server maintains time series information of all individual local models using a moving averaging method and forecast their movements when some clients are dropped. The proposed method is applied to several real-world regression problems and evaluated empirically.

**Strengths:**

1. This paper introduces an interesting concept of gradient estimation, digital twin-based scheme.
2. The background is well defined and clearly explained with example figures.
3. The experiments are conducted using real-world datasets rather than popularly used small-scale benchmarks.

**Weaknesses:**

I appreciate the high-level idea of forecasting the missing clients’ updates and using them to mitigate the adverse impact of client dropouts. However, the proposed method lacks both soundness and novelty.

**Comments regarding the proposed ideas**
1. FedDTW leverages *weighted smoothing forecasting (WSF)8 to estimate the local updates of clients that drop out during the FL process (as described at line 13 in Algorithm 5). Although this is the key idea of the paper, it is not well substantiated. My question is why WSF is expected to forecast local updates with sufficiently high quality.

2. $\Delta ew^t$ is not clearly defined in the main text. What is it? Why should it be subtracted from the current model parameter $w^t$?

3. The smoothing factor $\alpha$ is fixed to 0.8, however, it is not explained why that specific value should be used.

4. The authors raise the issue of potential bias in updates caused by client dropouts, but they do not discuss how it is affected by the proposed update forecasting method. This bias primarily stems from unexpected dropouts, therefore, the discrepancy between the dropped updates and the forecasted ones should still introduce bias. This gap, as well as the impact of WSF on this bias, should be carefully analyzed. Otherwise, the effectiveness of FedDTW cannot be properly evaluated.

5. The whole paper talks about *digital twin*, yet the main idea appears largely unrelated to actual digital twin systems. Technically, the proposed approach functions more as a local update estimation scheme. The authors should clarify which aspects of FedDTW are genuinely connected to the notion of a digital twin and justify the use of this terminology.

6. The proposed method is evaluated only under limited settings. Model accuracy alone cannot justify the authors’ claims, as the effectiveness of the method has not been theoretically explained. For example, why is the validation error reduced by forecasting updates with WSF? Is WSF expected to outperform other moving average–based forecasting techniques across various scenarios? If so, why? The authors should place greater emphasis on explaining why the key idea serves as the most effective solution.

7. There is no comparison with other methods. Authors can easily find gradient estimation methods [1,2,3] which can simply replace WSF in Algorithm 5. Gradient reusing methods are also tightly related to the moving averaging [4,5]. Without fairly extensive comparisons with them, the proposed idea becomes just a simple application of WSF in FL which mimics full-partificpation of clients.

8. In order to keep the moving average of previous updates, server may consume a large amount of extra memory space. The memory footprint even increases as the number of participating clients increases. This limitation has not been discussed nor evaluated in experiments. I suggest at least theoretically analyzing the memory requirements.

[1] Mohamed et al., Monte Carlo Gradient Estimation in Machine Learning, JMLR, 2020.

[2] Paulus et al., Gradient Estimation with Stochastic Softmax Tricks, NeurIPS, 2020.

[3] Vieol et al., Unbiased Gradient Estimation in Unrolled Computation Graphs with Persistent Evolution Strategies, ICML 2021.

[4] Azam et al., Recycling Model Updates in Federated Learning: Are Gradient Subspaces Low-Rank?, ICLR, 2022.

[5] Lee et al., Layer-wise Update Aggregation with Recycling for Communication-Efficient Federated Learning, NeurIPS, 2025.

**Comments regarding presentation quality**

1. I do not think Figure 1 is necessary. It occupies a critical position in the Introduction, yet it merely illustrates an intuitive scenario where some clients drop out during a communication round. In general, a figure placed in such a position is expected to convey the key motivation or main idea of the paper. I suggest replacing it with digital twin-related schematic illustration for example.

2. Figure 2 uses acronyms such as RCD, VCPR, and DU. To make each figure self-contained, I suggest defining them in the caption. Since they are defined elsewhere (in the following algorithms), readers have to search for their meanings.

3. Algorithm 1~4 are not *algorithms* but scenario settings. I do not think they should take up half a page. Author can simply summarize their behaviors as a couple of bullets.

**Questions:**

My questions are included in the above Weaknesses section. Please carefully address them to strengthen the paper.

---

### Official Review · Reviewer_on42 · 2025-10-29

**Soundness:** 2
**Presentation:** 2
**Contribution:** 2
**Rating:** 2
**Confidence:** 4

**Summary:**

The paper proposes FedDTW, a server-side framework that maintains digital twin replicas of clients’ models in FL. The authors claim that this framework is able to
++mitigate biases caused by unreliable client connectivity
++improve global model convergence under various dropout scenarios
++work across multiple backbone models
++outperforms FedAvg in RMSE and MAE across multiple real-world time-series datasets

**Strengths:**

++The approach is computationally light and does not require changes to client-side computation
++Evaluation across four datasets, four FL instability scenarios
++Public code and data availability

**Weaknesses:**

++The paper lacks theoretical analysis of convergence guarantees or bias correction under the introduced forecasting mechanisms. The forecast-based aggregation is intuitive but not formally justified.
++Using MAF and WSF to predict neural weights is simplistic and may not generalize. There is no evidence these models capture complex weight dynamics in non-stationary settings.
++The main comparison is only with FedAvg. It omits stronger baselines like FedVARP, FedAR, FedStale, or Mimic, which address similar problems.
++The improvement percentages are sometimes unclear. Statistical significance tests are missing.
++The cost of maintaining per-client digital twins and forecasting large weight vectors is not discussed.
++While the paper claims FedDTW handles non-IID data, there is no dedicated experiment isolating heterogeneity effects or ablation studies examining different levels of non-IID-ness.

**Questions:**

Please address all the aforementioned weaknesses

---

### Official Review · Reviewer_icKA · 2025-10-30

**Soundness:** 3
**Presentation:** 3
**Contribution:** 2
**Rating:** 4
**Confidence:** 3

**Summary:**

This paper addresses the problem of client dropouts in cross-device Federated Learning (FL), which can lead to biased global updates and slower convergence. The authors introduce Federated Digital Twin Weighting (FedDTW), a lightweight, server-side mechanism. FedDTW maintains a "digital twin" of each client's model, which is essentially a historical trajectory of its weights. When a client fails to participate in a round, the server uses this history to forecast the client's likely current parameters (using simple methods like Moving Average Forecasting or Weighted Smoothing Forecasting). This forecasted update is then used in the global aggregation step to "fill in" the missing client's contribution. The authors evaluate FedDTW across four realistic participation scenarios (e.g., random dropout, network partitions) and four time-series datasets, showing that FedDTW achieves a lower RMSE (closer to a full-participation reference) than standard FedAvg.

**Strengths:**

The paper tackles a significant and practical problem in real-world cross-device FL.

FedDTW is a lightweight, server-side-only mechanism. This is a major advantage as it requires no changes to the client-side training protocol, making it easy to deploy.

The evaluation across four different and realistic dropout scenarios (RCD, VCPR, NP, DU) on multiple datasets is comprehensive and demonstrates the robustness of the proposed method in various settings.

The method consistently outperforms the standard FedAvg baseline, showing its effectiveness in mitigating the bias introduced by partial participation.

**Weaknesses:**

The core weakness is the limited conceptual novelty. The idea of using historical client information to handle dropouts is established (e.g., FedVARP, FedStale, or even server-side momentum). The paper's main contribution is applying simple forecasting instead of re-using stale updates. It needs to more clearly differentiate itself from these prior works, especially with an experimental comparison.

The paper claims to be for "cross-device settings," which often imply millions of clients. However, FedDTW requires the server to maintain a "digital twin" (historical state) and perform forecasting for every client in the federation. The memory and computational overhead of this on the server is not discussed but seems non-trivial and would scale linearly (or worse, if forecasting is complex) with the total number of clients, N. The experiments seem to use a small number of clients (e.g., 3-50, based on dataset descriptions), which doesn't address this scalability question.

The paper uses very simple forecasting models (MAF, WSF). While this is praised as "lightweight," it's questionable whether this is sufficient for the high-dimensional, non-stationary weight trajectories of large neural networks. What happens if a client is unavailable for many (e.g., 10, 20, 50) consecutive rounds? The forecast error from such a simple model would likely compound, potentially leading to model divergence.

**Questions:**

Could the authors provide a direct experimental comparison against closely related methods like FedVARP (re-using the last received update) or FedStale? This would help quantify the actual benefit of forecasting versus just re-using stale data.

What is the server-side memory and computational overhead of FedDTW? How does the approach scale as the total number of clients (N) grows from the tens (used in the experiments) to the millions (common in cross-device FL)?

How does FedDTW's performance degrade as a client's dropout period increases? Can the simple MAF/WSF models provide a useful forecast if a client is offline for 20 or 50 rounds, or does the error compound and harm the global model?

---

### Note · Authors · 2025-11-30

I have read and agree with the venue's withdrawal policy on behalf of myself and my co-authors.